# SensiMix: Sensitivity-Aware 8-bit index & 1-bit value mixed precision quantization for BERT compression

**Tairen Piao, Ikhyun Cho, U. Kang**  *

Seoul National University, Seoul, Republic of Korea

* ukang@snu.ac.kr

## Abstract

Given a pre-trained BERT, how can we compress it to a fast and lightweight one while maintaining its accuracy? Pre-training language model, such as BERT, is effective for improving the performance of natural language processing (NLP) tasks. However, heavy models like BERT have problems of large memory cost and long inference time. In this paper, we propose SᴇɴsɪMɪx (Sensitivity-Aware Mixed Precision Quantization), a novel quantization-based BERT compression method that considers the sensitivity of different modules of BERT. SᴇɴsɪMɪx effectively applies 8-bit index quantization and 1-bit value quantization to the sensitive and insensitive parts of BERT, maximizing the compression rate while minimizing the accuracy drop. We also propose three novel 1-bit training methods to minimize the accuracy drop: Absolute Binary Weight Regularization, Prioritized Training, and Inverse Layer-wise Fine-tuning. Moreover, for fast inference, we apply FP16 general matrix multiplication (GEMM) and XNOR-Count GEMM for 8-bit and 1-bit quantization parts of the model, respectively. Experiments on four GLUE downstream tasks show that SᴇɴsɪMɪx compresses the original BERT model to an equally effective but lightweight one, reducing the model size by a factor of 8× and shrinking the inference time by around 80% without noticeable accuracy drop.

## Introduction

Given a pre-trained BERT, how can we compress it to a fast and lightweight one? Pre-training language models, such as BERT [1], RoBERTa [2], and ERNIE 2.0 [3], have been shown to be effective for improving many natural language processing (NLP) tasks, such as language inference, named entity recognition, and question answering.

However, these models usually have an extremely large number of parameters, which leads to high training cost and long inference time. For example, BERT-base [1] has 12 layers and 110 million parameters, and training BERT-base from scratch typically takes about four days on 4 to 16 Cloud TPUs. Even fine-tuning on downstream tasks may take several hours to finish on a typical GPU like NVIDIA RTX 2080Ti. Moreover, Kovaleva et al. [4] demonstrate there is redundancy in BERT. Therefore, it is crucial to reduce the large model size, long inference time, and computational overhead of BERT while retaining its accuracy.

**Data Availability Statement:** The data underlying this study have been uploaded to GitHub and are accessible using the following link: https://github.com/snudatalab/SensiMix.

**Funding:** This work was supported by Institute of Information & Communications Technology Planning & Evaluation (IITP) grant funded by the Korea government (MSIT) ([grant number No.2020-0-00894, Flexible and Efficient Model Compression Method for Various Applications and Environments], [grant number No.2017-0-01772, Development of QA systems for Video Story Understanding to pass the Video Turing Test], [grant number No.2021-0-02068, Artificial Intelligence Innovation Hub (Artificial Intelligence Institute, Seoul National University)], and [grant number No.2021-0-01343, Artificial Intelligence Graduate School Program (Seoul National University)]). The Institute of Engineering Research and ICT at Seoul National University provided research facilities for this work. The funders had no role in study design, data collection and analysis, decision to publish, or preparation of the manuscript.

**Competing interests:** The authors have declared that no competing interests exist.

Model compression has been widely studied in recent years [5–11] due to the increase of the model size, and there are several methods shown to be effective on BERT compression [12–17], such as knowledge distillation (KD)-based, pruning-based, parameter sharing-based, and quantization-based ones. Nevertheless, these methods have several limitations. First, KD-based methods do not give high compression rates. For example, DistilBERT-base [12] compresses the model size only to 61.2% of the original model. Second, there is a huge accuracy degradation when compressing a large proportion of BERT. For example, a pruning-based method [13] reduces the model size considerably, but it has a significant accuracy drop. Third, parameter sharing-based methods (e.g., ALBERT [14]) successfully reduce the model size, but fail to decrease the inference time. Furthermore, various approaches such as modifying pre-training task of BERT [18], designing a hardware-friendly version of BERT [19], applying a plug-and-play approach to efficiently reuse the parameters of BERT [20] have also been recently proposed to increase the computational efficiency of BERT from different perspectives.

Among these various compression methods, in this paper we focus on the field of quantization due to its superior capability of compressing models. However, previous quantization-based methods (e.g., Q8BERT [15]) quantize BERT directly without considering the sensitivity of different modules of BERT. We define the sensitivity of a module to be the degree of change in the accuracy by the change of the module, at the same compression rate. When compressing different modules with the same compression rate, those causing severe accuracy drop are more sensitive. There are two problems for methods that do not consider the sensitivity of modules. One is that these methods lead to an accuracy drop caused by compressing too many sensitive parameters; the other is that insensitive parameters are not optimally compressed.

In this paper, we propose SᴇɴꜱɪMɪx, a novel quantization-based BERT compression method. We improve the efficiency of BERT compression in the following three-aspect: model size, accuracy, and inference speed. We decrease the size of BERT by our proposed sensitivity-aware mixed precision quantization, which improves the previous quantization approaches by choosing target compression ratios based on the sensitivity of modules in BERT. We demonstrate that the encoders close to the input layer are more sensitive than those near to the output layer in BERT, and Self-Attention layer is more sensitive than feed-forward network (FFN) in an encoder. Hence, SᴇɴꜱɪMɪx quantizes these sensitive parts to 8-bit and the remaining parts to 1-bit. For the 8-bit quantization, we introduce 8-bit index quantization to reduce the model size while retaining the accuracy by using 8-bit indices, minimum weight, and maximum weight to efficiently represent all weights of each layer. We initialize the SᴇɴꜱɪMɪx model using a pre-trained BERT model and fine-tune the SᴇɴꜱɪMɪx model on downstream tasks based on the sensitivity-aware quantization strategy.

We then improve the accuracy of SᴇɴꜱɪMɪx by our proposed three training methods for 1-bit quantization. First, we propose Absolute Binary Weight Regularization (ABWR), which makes the absolute value of each 32-bit floating point (FP32) full-precision weight close to 1 in the training phase to overcome the loss of model precision. Second, we propose Prioritized Training (PT). PT lets FP32 full-precision weights learn the binary input features before binarizing both input and weights to alleviate the accuracy drop caused from the lack of knowledge of binary input features by the initial full-precision weights. Third, we introduce Inverse Layer-wise Fine-tuning (ILF). ILF gradually increases the proportion of 1-bit parameters during training, which alleviates the accuracy drop caused by quantizing too many parameters to 1-bit at once. In the inference phase, SᴇɴꜱɪMɪx applies FP16 general matrix multiplication (GEMM) to the 8-bit parts of the model and XNOR-Count GEMM to the 1-bit parts to achieve a fast inference speed. Fig 1 shows that SᴇɴꜱɪMɪx shows the best trade-off between accuracy, model size, and inference time.

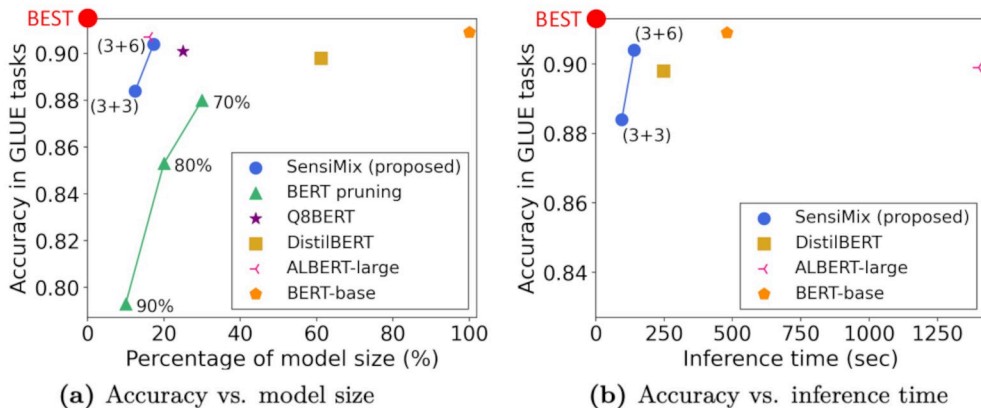

**Fig 1. SensiMix shows the best trade-off between accuracy, model size, and inference time among the competitors.** We report the average accuracy of four GLUE tasks (QQP, QNLI, SST-2, and MRPC).

Our main contributions are as follows:

- **Sensitivity-aware mixed precision quantization.** We propose SᴇɴꜱɪMɪx, a BERT compression method which exploits mixed precision quantization considering the sensitivity of different modules. SᴇɴꜱɪMɪx quantizes sensitive parts of the model using 8-bit index quantization and insensitive parts using standard 1-bit quantization, to achieve both high compression rate and accuracy.

- **Training methods for improving accuracy.** We propose Absolute Binary Weight Regularization, Prioritized Training, and Inverse Layer-wise Fine-tuning for training the 1-bit parts of the model, which alleviate the accuracy drop caused by applying the 1-bit quantization.

- **Inference strategy.** We apply FP16 GEMM to the 8-bit parts of the model and XNOR-Count GEMM to the 1-bit parts for achieving a fast inference speed.

- **Experiments.** We conduct the experiments on four GLUE downstream tasks. Experiments show that SᴇɴꜱɪMɪx compresses BERT 8× in terms of model size and gives 5× faster inference speed. Ablation study shows our three 1-bit training methods ABWR, PT, and ILF improve the average accuracy of SᴇɴꜱɪMɪx by 1.1%, 1, 4%, and 1.4%, respectively, compared to without applying them.

In the rest of this paper, we first introduce the related works and describe our proposed method. Then, we experimentally evaluate the performance of SᴇɴꜱɪMɪx and its competitors. The code of SᴇɴꜱɪMɪx is available at https://github.com/snudatalab/SensiMix. Table 1 shows the symbols used in this paper.

## Related work

### BERT

BERT [1] is a pre-trained language model which has achieved state-of-the-art performance on many downstream natural language processing tasks. The model consists of a WordPiece embedding layer, several Transformer [21] encoder layers, and a task-dependent classifier. The BERT model is trained with a large corpus data on masked language modeling and next sentence prediction tasks. The BERT-base model showed great success in a variety of NLP

**Table 1. Table of symbols.**

| Symbol | Definition |
|:---:|:---|
| $a_{fp}$ | Full-precision activation |
| $a_b$ | Binary activation |
| $w_{fp}$ | Full-precision weight |
| $w_b$ | Binary weight |
| $W_{fp}^l$ | Full-precision weight matrix of the layer $l$ |
| $W_b^l$ | Binary weight matrix of the layer $l$ |
| $W_q^l$ | Quantized 8-bit weight matrix of the layer $l$ |
| $W_{dq}^l$ | De-quantized FP32 weight matrix of the layer $l$ |
| $L$ | Loss function |
| $\gamma$ | Strength of the ABWR regularization term |
| $O_{ij}$ | $(i, j)^{th}$ element of the output matrix |

tasks, but it has 110M parameters which has problems of huge memory cost and long inference time. Hence, there has been a growing interest in compressing BERT to a tiny one through various methods such as pruning, parameter sharing, knowledge distillation, and quantization.

## Network pruning

Network pruning is a technique to reduce the model size by removing the weights of a deep neural network, and it has been widely studied in recent years [5, 22, 23]. Based on the assumption that deep neural networks have redundancy, and many parameters in a deep neural network are unimportant or unnecessary, network pruning is used to remove unimportant parameters. In this way, pruning methods increase the sparsity of the parameters significantly. After pruning, the sparse model requires less space, by using compressed sparse row format (CSR) or compressed sparse column (CSC) format.

**BERT pruning.** Gordon et al. [13] explore the effectiveness of weight pruning on BERT and figure out how pruning during pre-training affects the accuracy of the model. The authors find that pruning affects the performance of BERT in three broad regimes. Low levels of pruning (30-40%) have a negligible effect on pre-training loss and transferring the knowledge to downstream tasks. Medium levels of pruning (40-70%) increase the pre-training loss and hinder useful pre-training information from being transferred to downstream tasks. High levels of pruning (70-90%) additionally prevent models from fitting to downstream tasks, leading to further accuracy degradation. Experiments show that pruning equal to or less than 60% parameters of BERT affects the accuracy of the model marginally on GLUE benchmark [24] tasks, but pruning more than 70% starts to considerably influence the accuracy. On top of that, the method has a significant disadvantage: it is difficult to reduce the inference time without special hardware [25]. Compared to pruning methods, SENSIMIX reduces both the number of layers and size of parameters without pruning any parameter in the model, retaining higher accuracy and gaining faster inference speed.

## Parameter sharing

Parameter sharing is a widely used technique to reduce the number of parameters of a model [6, 26]. The basic idea of parameter sharing is to use the same parameters across different

layers in the model. However, despite the desirable reduction in the number of parameters, it generally suffers from a considerable accuracy drop.

**ALBERT.** ALBERT [14] is a novel lightweight version of BERT using the parameter sharing-based method. The cross-layer parameter sharing method is one of the main ideas of this model, which is to share the parameters across the layers of BERT. The authors use various parameter sharing methods such as all-shared, shared-attention, and shared-FFN methods. All these sharing schemes reduce the accuracy of the model, but the authors scaled up the size of the model after sharing parameters not only to recover but exceed the original accuracy of the model. However, ALBERT fails to reduce the inference time because the input still needs to pass through 12 or more layers, which is a critical disadvantage since inference time is a significant factor in model compression. Compared to ALBERT, SᴇɴsɪMɪx compresses the model via reducing the number of layers and quantizing the parameters, leading to much faster inference speed.

### Knowledge distillation

Knowledge distillation (KD) [7] aims to train a compact or smaller *student* model to approximate the function learned by a large and complex *teacher* model. Recently, KD has become one of the main techniques for model compression, by training a small student model to imitate the soft output labels of a large teacher model. Romero et al. [27] demonstrate that intermediate representations learned by the large model could serve as hints to improve the training process and the final performance of the student model. Moreover, Liu et al. [28] apply KD to transfer knowledge from ensemble models to improve the performance of a single model on natural-language understanding (NLU) tasks.

**DistilBERT.** DistilBERT [12] explores the problem of compressing BERT by applying KD. DistilBERT has the same general architecture as BERT, but the number of encoder layers is reduced by a factor of 2. DistilBERT applies KD on a student model during pre-training, allowing the student model to learn the soft label of the teacher's outputs. DistilBERT is trained on huge batches leveraging gradient accumulation using dynamic masking and without the next sentence prediction task. Experiments show that DistilBERT achieves a good result on the GLUE benchmark with only a slight accuracy drop compared to BERT-base. However, the downside of DistilBERT is that even though it reduces half of the encoder layers, the embedding layer consists of about 21% of the total parameters, making the model size to only 61.2% of the original. Compared to DistilBERT, SᴇɴsɪMɪx additionally compresses the parameters by quantization, reducing the model size and improving the inference speed.

### Network quantization

Network quantization uses smaller bit-width integers to represent and compress parameters of deep neural networks. A typical deep learning model uses 32-bit floating point (FP32) format for its parameters. [8, 29] demonstrate that weights and activations can be represented using 8-bit numbers without significant accuracy drop. The use of even lower bit-widths such as 4, 2, and 1-bits has also shown remarkable progress [30–33]. For example, binarized neural network (BNN) [32] uses 1-bit for each parameter to save storage and reduce computation. However, the quantized model, such as BNN, causes severe loss of precision and accuracy drop.

**Q8BERT.** Q8BERT [15] applies 8-bit quantization-aware training during the fine-tuning process of BERT. In forward propagation, Q8BERT first quantizes the activation and weight matrices to INT8 format by multiplying two scaling factors of the two matrices, performs INT8 GEMM, which multiplies and accumulates two INT8 matrices to an INT32 matrix, and then de-quantizes the INT32 matrix to an FP32 matrix by dividing the scaling factors of the

weight and activation matrices. In backward propagation, Q8BERT uses the 8-bit clip function to approximate the 8-bit round function for training the model. Q8BERT reduces the model size by a factor of 4 and maintains almost the same accuracy as the BERT with FP32 precision in eight different NLP tasks. However, Q8BERT does not consider the sensitivity of layers in BERT and does not apply mixed precision quantization strategy; on the other hand, our SENSI-MIX applies the mixed precision quantization considering the sensitivity and provides a superior performance compared to Q8BERT.

**KDLSQ-BERT & K-means quantization.**    Jin et al. [33] propose KDLSQ-BERT, a framework to combine quantization with knowledge distillation using BERT. KDLSQ-BERT adopts learned step size quantization (LSQ; [34]), which is a variant of the original quantization-aware training, which has been shown to be effective in computer vision. Different from the ordinary quantization-aware training, LSQ additionally learns the scale factor for each weight and activation during the training process. By applying LSQ along with KD on BERT, KDSLQ-BERT shows decent trade-off between performance and memory footprint. Also, Zhao et al. [35] propose a variant of quantization-aware training by adopting the idea of k-means clustering into quantization. They show that this k-means clustering-based variant is comparable to the original quantization-aware training on BERT-based models.

In our paper, since considering every variant of quantization method is impractical, we focus on the linear and symmetric quantization-aware training, which is the most basic and universal quantization scheme, to verify the effectiveness of our SensiMix. Since SENSIMIX proves to be effective on this most fundamental quantization scheme, we believe SENSIMIX can be deployed for those variant settings as well. In addition, the three proposed training methods ABWR, PT, and ILF of SENSIMIX are orthogonal techniques making them complementary to other mixed and/or low-precision quantization methods.

**I-BERT.**    Kim et al. [36] present I-BERT which quantizes the entire inference of BERT-based models with integer-only arithmetics, avoiding any floating point calculations. The major merit of integer-only inference is that it can benefit from faster inference speed by using a family of specialized hardware that supports efficient integer computations (e.g. Turing Tensor Cores, ARM Cortex-M, etc.). Specifically, I-BERT approximates GELU and Softmax functions with second-order polynomials and LayerNorm with a fair number of iterations of integer arithmetic, which can then be evaluated with integer arithmetic-only hardware devices.

Different from I-BERT, SENSIMIX focuses on simulated quantization, which includes both quantization and de-quantization steps in inference, rather than restricting the setting to integer-only. This results in higher flexibility and applicability of SENSIMIX.

**EvoQ & Lee et al. [37].**    Yuan et al. [38] present EvoQ, a post-training quantization method which applies tournament selction, a classical evolutionary algorithm, to find the best quantization policy. Their sensitivity metric measures the output and intermediate layer differences between the quantized model and the full-precision model. Using this sensitivity metric along with the tournament selection algorithm, EvoQ improves the search efficiency for finding the best quantization policy.

Similarly, Lee et al. [37] also propose a metric to measure the layers' sensitivity to quantization. They measure the sensitivity of a specific layer by considering the effect of quantization on both the task loss (final output) and other intermediate layers using the concept of gradient perturbation. They also develop a neural network-agnostic data generation method to improve the quality of the quantized network.

These works are similar to ours in the sense that they attempt to measure the sensitivity of each layer to quantization. However, both of them are specialized to CNN-based models and thus not directly applicable to BERT.

## Proposed method

We propose SENSIMIX, a sensitivity-aware mixed precision quantization method for BERT compression. We first provide a brief overview of our method. Then, we describe the sensitivity-aware mixed precision quantization strategy. After introducing the 8-bit index quantization and the 1-bit value quantization, we describe how to perform inference of our model.

### Overview

Our goal is to compress BERT to a fast and lightweight model. We concentrate on the following challenges for the goal.

1. **Accuracy degradation caused by compression.** Many existing BERT compression methods lose the accuracy after compression. For example, Gordon et al. [13] reduce the model size of BERT considerably, but the accuracy is degraded significantly. How can we compress BERT to a lightweight one while keeping its accuracy?

2. **Challenges of 1-bit quantization.** We apply 1-bit quantization to insensitive parts of the model, but there are several challenges when applying 1-bit quantization. First, quantizing the original full-precision weights to 1 or -1 leads to a huge precision loss. Second, binarizing both weights and activations at the beginning of the training phase causes an accuracy drop due to the lack of binary input feature knowledge learned by the pre-trained FP32 precision model. Third, models that quantize too many parameters to 1-bit at once are hard to be trained. How can we overcome these challenges and improve the accuracy of the quantized model?

3. **Limitation of inference speed.** Many BERT compression techniques do not improve the inference speed. For example, ALBERT-large [14] has about 5× fewer parameters compared to BERT-base, but the inference time is 3× longer. How can we achieve a fast inference speed while maintaining a small model size and similar accuracy?

We address the mentioned challenges with the following ideas:

1. **Sensitivity-aware mixed precision quantization.** SENSIMIX quantizes the sensitive parts of BERT using 8-bit index quantization and insensitive parts using 1-bit value quantization, which maximizes the compression rate and minimizes the accuracy drop caused by quantizing sensitive modules to low bits.

2. **Training methods for 1-bit quantization.** We propose three training methods for the 1-bit quantization: Absolute Binary Weight Regularization (ABWR), Prioritized Training (PT), and Inverse Layer-wise Fine-tuning (ILF). They overcome the challenges of the conventional 1-bit quantization method mentioned above, improving the accuracy of the model.

3. **Fast matrix multiplications for inference.** We apply FP16 general matrix multiplication (GEMM) and XNOR-Count GEMM to replace the original GEMM for the 8-bit and the 1-bit quantization parts of the model to achieve fast inference speed.

Fig 2 compares the architecture of SENSIMIX to that of the original BERT. SENSIMIX effectively applies 8-bit index quantization and the 1-bit value quantization to sensitive and insensitive modules of BERT, respectively.

### Sensitivity-Aware mixed precision quantization strategy

Our goal is to find an effective quantization strategy to compress parameters in BERT while maintaining the accuracy of the model. BERT consists of a WordPiece embedding layer and

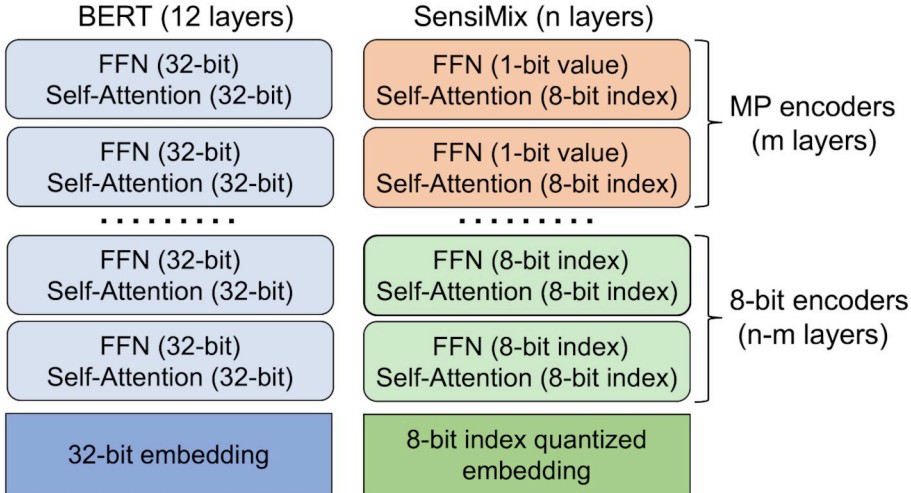

**Fig 2. Overview of SensiMix.** SENSIMIX applies 1-bit value quantization to insensitive feed-forward network (FFN) near the output layer, and applies 8-bit index quantization to remaining sensitive parts.

12 Transformer [21] encoder layers, where each encoder layer is composed of a Self-Attention layer and a feed-forward network (FFN). There is a classifier after the last encoder layer. Previous BERT compression methods do not consider the sensitivity of different modules of BERT, which leads to two problems. First, they cause an accuracy drop by compressing too many sensitive parameters of the model. Second, they do not compress parameters efficiently since the insensitive parts have not been optimally compressed.

**Algorithm 1:** SensiMix: Sensitivity-aware mixed precision quantization.

**Input:** $l$-th layer's input matrix $I^l$, full-precision weight matrix of the $l$-th layer $W_{fp}^l$, the number of encoder layers $N$, index of the $l$-th layer $l$, and the activation function $Act$

**Output:** The updated weight matrix of l-th layer $W_{updated}^l$

**1** Forward propagation:
**2** $l \leftarrow 1$
**3 for** $l \leq N$ **do**
**4** **if** $l$-th layer is the 8-bit encoder **then**
**5** $W_q^l \leftarrow W_{fp}^l$ // 8-bit index quantization (Eq 1)
**6** $W_{dq}^l \leftarrow W_q^l$ // de-quantization, (Eq 2)
**7** **else if** $l$-th layer is the MP encoder **then**
**8** **if** fully connected layer **then**
**9** $W_b^l \leftarrow W_{fp}^l$ // 1-bit value quantization (Eq 5)
**10** **else if** self-attention layer **then**
**11** $W_q^l \leftarrow W_{fp}^l$ // 8-bit index quantization (Eq 1)
**12** $W_{dq}^l \leftarrow W_q^l$ // de-quantization (Eq 2)
**13** $Act(W_{dq} \cdot I^l)$ (standard forward propagation)
**14** $l \leftarrow l + 1$
**15** Backward propagation:
**16** $l \leftarrow N$
**17 for** $l \geq 1$ **do**
**18** **if** $l$-th layer is the 8-bit encoder **then**
**19** Use the 8-bit clip function to replace the round function when computing the gradient of $W_{fp}^l$ // (Eq 4)

**20** Update $W_{fp}^l$ to $W_{updated}^l$

**21** **else if** *l-th layer is the MP encoder* **then**

**22** Use the 1-bit clip function to replace the round function when computing the gradient of $W^l$ // (Eq 6)

**23** Update $W_{fp}^l$ to $W_{updated}^l$

**24** *l ← l - 1*

To alleviate these problems, we pay attention to the sensitivity of different modules of the model. First, we discover that the Self-Attention layer is more important than FFN in an encoder. The Self-Attention layer calculates the relations between input word embeddings, which plays a crucial role in improving the accuracy of BERT. Moreover [39, 40], demonstrate that Self-Attention distributions of pre-trained language models capture a rich hierarchy of linguistic information, which reveals the importance of the Self-Attention layer. We also discover that the encoders close to the input layer are more sensitive than the encoders near the output layer in BERT. The encoders near the input layer extract important low-level features from the input embeddings, which are crucial for the model accuracy. Lin et al. [41] show that the layers close to the input layer have the most important information about linear word order, which signifies the importance of these layers. We verify these claims in the experiments part.

**Quantization strategy.** There are many choices of the number of bits to quantize the full-precision model. As we lower the number of bits used, we can save more memory, but the model accuracy also falls significantly. We basically want to compress the model to 1-bit because it provides good compression rate and fast inference speed, but applying 1-bit value quantization to all modules causes severe accuracy degradation. On the other hand, 8-bit index quantization reduces the memory storage needed to one-fourth while maintaining most of the accuracy compared to the FP32 model. Hence, we apply 1-bit value quantization to insensitive parts of the model and 8-bit index quantization to the remaining parts.

Based on the motivation, we propose SENSIMIX which contains two types of encoders, 8-bit encoder and Mixed Precision (MP) encoder. An 8-bit encoder is composed of 8-bit index-quantized FFN and Self-Attention layer, and an MP encoder is composed of 1-bit value-quantized FFN and 8-bit index-quantized Self-Attention layer. Given *n* layers from a pre-trained BERT model, we use *m* MP encoders near the output layer, and *n* − *m* 8-bit encoders for remaining *n* − *m* layers. Note that if *n* is smaller than the total layers of the BERT model, we initialize the model using the lower layers of the BERT model. For example, we initialize the SENSIMIX (3+3) model using layers 1-6 of the BERT-base model. Additionally, we apply 8-bit index quantization to the embedding layer. We do not quantize the bias layers, the LayerNorm layers, and the classifier since they occupy only a small part of the model. One advantage of SENSIMIX is that it is more flexible than existing methods (e.g., Q8BERT) because the type of encoder, as well as the number of layers, is also flexible. The overall process of SENSIMIX is shown in Algorithm 1.

SENSIMIX is trained during the fine-tuning stage of BERT. We initialize the model using a pre-trained BERT model and fine-tune the SENSIMIX model on downstream tasks based on the above sensitivity-aware mixed precision quantization strategy.

## 8-bit index quantization

We describe our 8-bit index quantization method in detail. The main idea of 8-bit index quantization method is to transform and shrink the original 32-bit weight matrices into 8-bit matrices composed of integer indices, thereby reducing the memory needed to save the model weights. As we lower the number of bits used, we can save more memory, but the model accuracy also falls significantly. Therefore, an appropriate compromise is required, and we

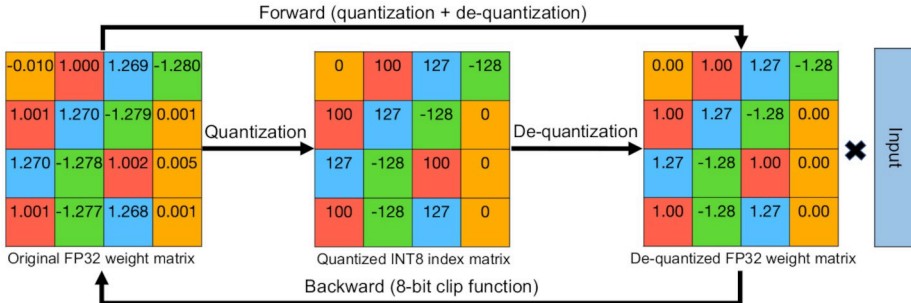

**Fig 3. The overall process of the 8-bit index quantization.** In the forward propagation, the original weights are first quantized to the 8-bit indices and then de-quantized back to FP32 weights, and in the backward propagation, we use the 8-bit clip function to replace the round function to train the model.

empirically choose 8-bit for our index quantization in SENSIMIX. Specifically, 8-bit index quantization reduces the memory storage needed to one fourth while maintaining most of accuracy in comparison to the FP32 version of the model. 8-bit index quantization consists of two steps: quantization and de-quantization. The quantization step quantizes FP32 values to 8-bit int values to reduce the memory requirement, and the de-quantization step de-quantizes the values back to 32-bit to make the output similar to the original FP32 model. In the following, we elaborate on the quantization step, the de-quantization step, and how we apply them in the training process. Fig 3 shows the overall process of the 8-bit index quantization.

**Quantization.** The quantization step reduces FP32 weight matrices into INT8 weight matrices. From an FP32 weight matrix, each FP32 weight is mapped to a number within [-128, 127] range (8-bit). We divide the minimum and the maximum weight range into 256 uniform intervals. Then for each weight in the matrix, if the value is in the $i^{th}$ interval, $i = 0, \ldots, 255$, then the integer ($i$-128) is assigned as its index. Through this quantization step, we map the FP32 weight matrix (the weight matrix of each layer) into an INT8 index matrix, which saves memory storage by a factor of 4. In addition to the mapped INT8 index matrix, we also store the minimum and maximum values of the original FP32 weight matrix for the later de-quantization step. For example, assume the original FP32 weight matrix consists of four numbers: -1.28, 0.005, 1.00, and 1.27. The range of the weights is [-1.28, 1.27] so the quantization unit is 0.01. Since -1.28, 0.005, 1.00, and 1.27 are in the $0^{th}$, $128^{th}$, $228^{th}$, and $255^{th}$ units, respectively, they are assigned indices -128, 0, 100, and 127, respectively. After the quantization step, the integer indices -128, 0, 100, 127 along with the minimum and maximum values -1.28 and 1.27 are stored. The formula of this quantization step is as follows:

$$W_q^l = round\left[\left(W_{fp}^l - min(W_{fp}^l)\right) \times \frac{127 - (-128)}{max(W_{fp}^l) - min(W_{fp}^l)} - 128\right], \tag{1}$$

where $W_q^l$ denotes the 8-bit index-quantized weight matrix of the $l^{th}$ layer, $W_{fp}^l$ denotes the FP32 full-precision weight matrix of the $l^{th}$ layer, $min(W_{fp}^l)$ and $max(W_{fp}^l)$ denote the minimum and maximum weights of the full-precision matrix $W_{fp}^l$, respectively.

**De-quantization.** In addition to the quantization step, a de-quantization step is required to preserve the accuracy of the original FP32 model. Since quantized index values differ much from the original FP32 weight values, directly using the quantized indices produces a very different output compared to the original FP32 model. Therefore, we use a step called de-quantization to map each quantized index back to an FP32 value similar to the original weight,

making the output similar to that of the FP32 model. The process of de-quantization is roughly a reverse process of quantization. We map the indices back to FP32 values based on the minimum and maximum weights we stored in the quantization step. Specifically, we divide the minimum and maximum weight range stored from the quantization step into 256 uniform units. Then for each index $j$ ($-128 \leq j \leq 127$) we assign the beginning FP32 value of the $(j + 128)^{th}$ unit. For example, from the former example of quantization, the minimum and maximum weight values that we stored are -1.28 and 1.27, respectively. Thus, the interval unit is 0.01 and the indices -128, 0, 100, and 127 are mapped back to -1.28, 0.00, 1.00, and 1.27, respectively, similar to the original weights -1.28, 0.005, 1.00, and 1.27. The formula of this de-quantization step is as follows:

$$W_{dq}^l = \left[ \left( W_q^l - (-128) \right) \times \frac{max\left( W_{fp}^l \right) - min\left( W_{fp}^l \right)}{127 - (-128)} \right] + min(W_{fp}^l), \tag{2}$$

where $W_{dq}^l$ denotes the 8-bit de-quantized weight matrix of the $l^{th}$ layer. The quantization step enables us to reduce the needed memory storage and the de-quantization step enables us to preserve the accuracy of the model.

**Training with 8-bit index quantization.**   Applying the 8-bit index quantization method on deep learning models requires slight modifications. We describe the required modifications of the training process.

In the forward propagation of the training process, we apply quantization and de-quantization on the weights of the model. We then use the de-quantized weight matrices instead of the original weight matrices to calculate the forward propagation. In the backward propagation, we update weights using standard gradient descent. However, the derivative of the round function is equal to zero in almost all ranges, making the weights incapable of being updated in gradient descent based training. To tackle this, we use the 8-bit clip function to approximate the round function to update the weights. The update rule and the 8-bit clip function are as follows:

$$W_{fp}^{l,t+1} = W_{fp}^{l,t} - \eta \frac{\partial L}{\partial W_{fp}^{l,t}} = W_{fp}^{l,t} - \eta \frac{\partial L}{\partial W_{dq}^{l,t}} \frac{\partial W_{dq}^{l,t}}{\partial W_{fp}^{l,t}} \tag{3}$$

$$round(x) \approx clip(x, -128, 127) = min(max(x, -128), 127), \tag{4}$$

where $L$ is the loss function of the model, superscripts $t$ and $t + 1$ represent the moments before and after weight updates, respectively. Note that in Eq 1, all original FP32 weight values are mapped into range [-128, 127] before input to the round function, which means all FP32 weights will be updated. After the training is done, we only store the quantized 8-bit index weight matrices and the maximum and minimum weights of each matrix, which reduces the model size by a factor of 4.

## 1-bit value quantization and additional techniques

We apply 1-bit value quantization instead of 8-bit index quantization for the *insensitive* weight matrices to further compress the model while maintaining most of its accuracy. Differently from the 8-bit index quantization, 1-bit value quantization does not have a de-quantization process; the weights are quantized to ±1 and directly propagated to the next layer of the model. For 1-bit value quantization, we adopt the approach [32] that reduces the model size by binarizing both weights and activations. The reason for binarizing both activations and weights is to apply XNOR-Count GEMM in the inference phase and boost the inference speed. We also

propose three additional 1-bit training methods called ABWR, PT, and ILF that minimize the accuracy drop caused by 1-bit value quantization. In the following, we first introduce the method of 1-bit value quantization in both forward and backward propagations and then introduce our three proposed 1-bit quantization-aware training methods.

**Forward propagation.** In the forward propagation, we first use the sign function to binarize weights and activations to either $+1$ or $-1$. We then use the binarized weights and activations to perform the standard forward process of the model. The mathematical formulas are as follows:

$$a_b = sign(a_{fp}) = \begin{cases} -1 & (a_{fp} \leq 0) \\ +1 & (a_{fp} > 0) \end{cases}, \quad w_b = sign(w_{fp}) = \begin{cases} -1 & (w_{fp} \leq 0) \\ +1 & (w_{fp} > 0) \end{cases}, \quad (5)$$

where $a_{fp}$, $w_{fp}$, $a_b$, and $w_b$ represent full-precision activation, full-precision weight, binary activation, and binary weight, respectively.

**Backward propagation.** We describe the backward propagation, showing how to update the binary weights $W_b^l$ in the $l^{th}$ layer. The derivative of the sign function is equal to zero in almost all ranges, making it incompatible with gradient descent-based training. To tackle this, 1-bit value quantization uses the 1-bit clip function to approximate the sign function to update $W_{fp}^l$:

$$W_{fp}^{l,t+1} = W_{fp}^{l,t} - \eta \frac{\partial L}{\partial W_{fp}^{l,t}} = W_{fp}^{l,t} - \eta \frac{\partial L}{\partial W_b^{l,t}} \frac{\partial W_b^{l,t}}{\partial W_{fp}^{l,t}} \quad (6)$$

$$clip(x, -1, 1) = min(max(x, -1), 1), \quad (7)$$

where $L$ is the loss function of the model, $\eta$ is the learning rate, and $W_b$ represents the binary weight matrix. By replacing the sign function with the 1-bit clip function, $\frac{\partial W_b^{l,t}(i,j)}{\partial W_{fp}^{l,t}(i,j)}$ is equal to 1 if $W_{fp}^{l,t} \in [-1, 1]$ and 0 otherwise. Note that Eq 6 is a basic training rule; other gradient descent-based update rules (e.g., Adam) can be used as well.

**Absolute Binary Weight Regularization (ABWR).** We propose Absolute Binary Weight Regularization (ABWR), a regularization method to reduce the precision loss caused by applying 1-bit quantization. Our goal is to reduce the precision loss by learning a new weight distribution that fits the 1-bit value quantization. Fig 4 shows the full-precision weight distributions of three binarized FFN in SENSIMIX (3+3) before and after applying ABWR on the QQP task. Note that 90% of the weights of pre-trained BERT are in the range of $[-0.07, 0.07]$, which are far from $\pm 1$. This is one of the main causes of the big drop in accuracy when 1-bit quantization is applied. To tackle this, we introduce a regularization term $L_R$ that restricts the absolute value of full-precision weights close to 1 in the training phase:

$$L_R = \frac{1}{2}(|w_{fp}| - 1)^2, \quad (8)$$

and the overall loss function is as follows:

$$L = L_B + \gamma \times L_R, \quad (9)$$

where $L_B$ denotes the original objective function of BERT, and $\gamma$ denotes the regularization coefficient.

The intuition of ABWR is to train the absolute value of the weights to become close to 1 in the first place, thereby minimizing the drop in accuracy when 1-bit quantization is applied. By

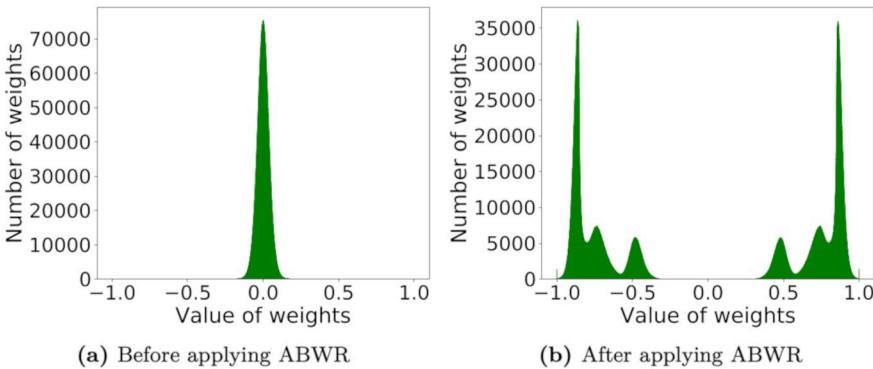

**Fig 4. Full-precision weight distributions of three binarized FFN in SensiMix (3+3) before and after applying ABWR on the QQP task.**

adding this regularizer in the loss function during training, the model reduces the precision loss caused by applying 1-bit quantization.

**Prioritized Training (PT).** We propose a new training method called Prioritized Training (PT) to overcome the difficulty in training the model with binary weights. Conventional 1-bit quantization methods binarize both input and weights from the beginning of the training, which means the full-precision weights have no chance to consider any binary input feature before applying 1-bit quantization. Hence, conventional methods do not give a good initial state of the model for 1-bit quantization because the initial full-precision weights have no knowledge about binary input. This is one of the causes that 1-bit quantized models are difficult to be trained. To tackle this problem, PT keeps the input binarized as in the conventional methods, but trains the weights in FP32 precision first, and then applies 1-bit quantization. The intuition behind PT is to provide a better initial state to the model for 1-bit quantization by training the full-precision weights to learn more binary input features before binarizing both input and weights. Due to this additional training, the model performs better than the traditional methods.

**Inverse Layer-wise Fine-tuning (ILF).** We propose Inverse Layer-wise Fine-tuning (ILF) to overcome the difficulty in training a SᴇɴsɪMɪx model that applies 1-bit value quantization to a large proportion of the model at once. Fig 5 shows the process of ILF. We observe that the

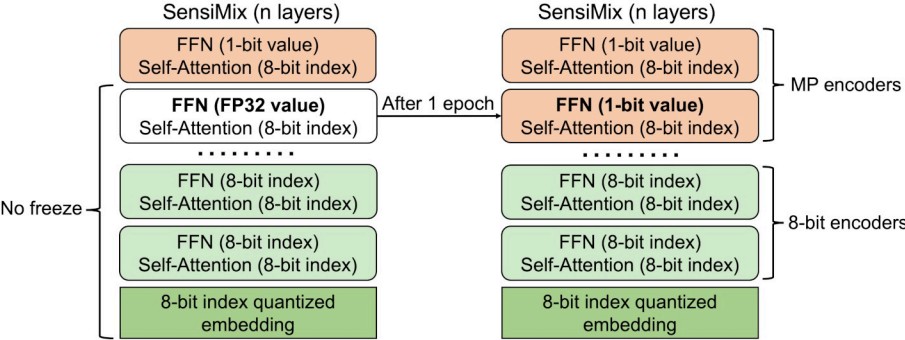

**Fig 5. Inverse Layer-wise Fine-tuning (ILF).** Given the model with $k$ MP encoder layers, ILF adds one more MP encoder layer on the bottom of it and fine-tune the model with $k + 1$ MP encoder layers.

models that deploy half or more MP encoder layers at once are hard to be trained. To tackle this problem, our proposed ILF first fine-tunes the model starting with one MP encoder layer, letting it converge. The first MP encoder is deployed on the top of the model (close to the output layer). After 1 training epoch, we iteratively perform the following procedure, for $k = 1, 2, \ldots, m - 1$: given the model with $k$ MP encoder layers, we add one more MP encoder layer on the bottom of the existing MP encoder layers and fine-tune the model with $k + 1$ MP encoder layers for 1 epoch. We stop the iteration when we have total $m$ MP encoder layers. Note that we do not freeze any parameter of the model in the process. ILF enables a more effective training of the final model with a gradual quantization approach.

## Model inference

We describe our implementation details for fast model inference.

**Inference for 8-bit.**   For 8-bit index quantization, we adopt FP16 GEMM to replace the FP32 GEMM. FP16 GEMM is a traditional GEMM where all elements in matrices are in FP16 precision. After training, we store the 8-bit index-quantized weight matrix, minimum weight, and maximum weight of each weight matrix. In the inference phase, we first de-quantize the saved INT8 indices back to FP32 precision weights following Eq (2). We then convert all the FP32 precision weights and activations to FP16 precision following the IEEE 754 standard [42]. After that, FP16 precision weights and activations are used instead of the FP32 version throughout the entire process. We also apply FP16 precision to the bias and LayerNorm layers to achieve a fast inference speed.

**Inference for 1-bit.**   For 1-bit value quantization, we apply XNOR-Count GEMM instead of the traditional GEMM to achieve an even faster inference speed. XNOR-Count GEMM is a fast matrix multiplication method for binary matrices, which is widely used in the inference phase of binarized neural networks. We explain how XNOR-Count GEMM works with an example. Consider multiplying two binary matrices (all elements are 1 or -1) $A$ and $B$. For computing the $(i, j)^{th}$ element of the output matrix, XNOR-Count GEMM first performs XNOR operation on the $i^{th}$ row vector of $A$ and the $j^{th}$ column vector of $B$ and yields an output vector. Then the method counts the number of 1 in the output vector, multiplies the result by 2, and deducts the dimension of the $i^{th}$ row vector of $A$ to get a scalar value. This value is equal to the $(i, j)^{th}$ element of the output matrix. The mathematical formula of XNOR-Count GEMM is as follows:

$$O_{ij} = 2 \times count(xnor(A_i, B_j)) - dim(A_i), \tag{10}$$

where $O_{ij}$ represents the $(i, j)^{th}$ element of the output matrix, $A_i$ represents the $i^{th}$ row vector of the binary matrix A, $B_j$ represents the $j^{th}$ column vector of the binary matrix B, and $dim(A_i)$ represents the dimension of the vector $A_i$.

However, it is difficult to directly apply this method in practice because modern deep learning frameworks do not support saving each value in 1-bit numerical format. Therefore, in the implementation step, we encode each binary matrix to an FP32 matrix where each FP32 element stores 32 binary values. We row-encode the left matrix and column-encode the right matrix for multiplication, and elements with values 1 and -1 are encoded to bits 1 and 0, respectively. We apply XNOR-Count GEMM on these FP32 matrices. For example, suppose we multiply two binary matrices $A$ and $B$ with sizes $(i, m)$ and $(m, j)$, respectively. We first encode them into FP32 precision matrices with sizes $(i, m/32)$ and $(m/32, j)$ where each FP32 number consists of 32 encoded 1-bit values. We then apply XNOR-Count GEMM to the

encoded matrices with bit-wise operations as follows:

$$O_{ij} = 2 \times \sum_{k=1}^{m/32} [popcount(xnor(A_{ik}, B_{kj}))] - 32 \times dim(A_i), \tag{11}$$

where *popcount* represents the bit-wise count operation. XNOR and popcount operations require fewer computational cost than the traditional GEMM, thus resulting in a faster inference speed.

**Theorem 1** (FLOPs of SENSIMIX). *Let $F_B$ be the FLOPs of the full-precision (32-bit) BERT-base model (12 layers). The FLOPs $F_S$ of SENSIMIX with n-layers and m-MP encoders is given by* $\Theta\left(\left(\frac{1}{24}n - \frac{5}{192}m\right)F_B\right)$.

*Proof.* We omit the bias, LayerNorm layers, and final classifier since they occupy a negligible part of the model. The Self-Attention layer occupies $\frac{1}{3}$ size in an encoder and FFN occupies the remaining $\frac{2}{3}$. The layer with the 8-bit index quantization uses the FP16 GEMM to make inference which has half of the FLOPs compared to FP32 GEMM. The layer with the 1-bit value quantization uses XNOR-Count GEMM to make inference which has 1/32 (both input and weight are 1-bit) of the FLOPs when using 32-bit popcount operations. Then,

$$F_S = \left((n - m) \times \frac{1}{2} \times \frac{F_B}{12}\right) + \left(\frac{1}{3}m \times \frac{1}{2} \times \frac{F_B}{12}\right) + \left(\frac{2}{3}m \times \frac{1}{32} \times \frac{F_B}{12}\right).$$

Thus,

$$F_S = \left(\frac{1}{24}n - \frac{5}{192}m\right)F_B.$$

We observe that the FLOPs of SENSIMIX is much less than that of BERT, and decreases when the number $m$ of MP encoders increases. The inference speed of SENSIMIX (3+3) is $\frac{6}{24} - \frac{15}{192} = 5.82$ times faster than BERT-base. Tables 2 and 3 show that SENSIMIX (3+3) gives 5.05 times faster inference time than BERT-base, which is close to the theoretical speed-up.

## Experiments

We run experiments to verify the effectiveness of our proposed method. Our goal is to answer the following questions.

- **Q1. Overall performance.** How does SENSIMIX perform compared to other methods in terms of accuracy, model size, and inference speed?

**Table 2. Overall performance of SensiMix compared to the competitors.** SENSIMIX achieves the smallest model size and the least inference time while maintaining the similar accuracy.

| Methods | Model Size (%) | Inference Time (Sec) | QQP (Acc) | QNLI (Acc) | SST-2 (Acc) | MRPC (F1) | Avg. |
|---|---|---|---|---|---|---|---|
| BERT-base | 100.0 | 480 | 90.8±0.3 | 90.5±0.3 | 92.1±0.2 | 90.2±0.5 | 90.9 |
| BERT pruning 70% | 30.0 | - | 88.6±0.5 | 86.4±0.2 | 89.5±0.3 | 87.3±0.1 | 88.0 |
| BERT pruning 80% | 20.0 | - | 86.9±0.2 | 82.3±0.6 | 87.1±0.6 | 84.8±0.4 | 85.3 |
| BERT pruning 90% | 10.0 | - | 81.1±0.2 | 72.5±0.7 | 80.3±0.3 | 83.1±0.3 | 79.3 |
| DistilBERT | 61.2 | 248 | 89.9±0.4 | 88.5±0.4 | 91.3±0.3 | 89.4±0.3 | 89.8 |
| ALBERT-large | 16.2 | 1401 | 90.9±0.4 | 90.1±0.3 | 91.8±0.1 | 90.1±0.1 | 90.7 |
| Q8BERT | 25.0 | - | 90.2±0.0 | 88.9±0.5 | 91.5±0.3 | 89.6±0.2 | 90.1 |
| SENSIMIX (3+6) | 17.3 | 140 | 90.4±0.3 | 89.0±0.5 | 92.0±0.1 | 90.0±0.1 | 90.4 |
| SENSIMIX (3+3) | 12.5 | 95 | 89.6±0.2 | 86.5±0.2 | 90.3±0.1 | 87.2±0.2 | 88.4 |

**Table 3. Inference time of SensiMix compared to the competitors.** SᴇɴꜱɪMɪx achieves the fastest inference speed compared to the competitors.

| Methods | QQP (sec) | QNLI (sec) | SST-2 (sec) | MRPC (sec) | Avg. (sec) |
|---|---|---|---|---|---|
| BERT-base | 1362±20 | 320±5 | 198±5 | 9±1 | 480 |
| DistilBERT | 712±8 | 168±4 | 101±3 | 6±1 | 248 |
| ALBERT-large | 4087±30 | 900±18 | 591±15 | 24±2 | 1401 |
| SᴇɴꜱɪMɪx (3+6) | 408±12 | 89±5 | 61±4 | 3±1 | 140 |
| **SensiMix (3+3)** | **272±10** | **66±3** | **40±3** | **2±1** | **95** |

- **Q2. Effectiveness of 1-bit training methods.** How do our ABWR, PT, and ILF affect the accuracy of SᴇɴꜱɪMɪx?

- **Q3. Sensitivity.** Which part of the model is more sensitive in an encoder, and which encoder is more sensitive in the model?

## Experimental settings

**Dataset.** We assess the performance of SᴇɴꜱɪMɪx on the General Language Understanding Evaluation (GLUE) benchmark [24]. GLUE is an evaluation system widely used in NLP. GLUE covers a wide range of tasks, including single-sentence tasks, paraphrase tasks, and inference tasks, to evaluate the model's overall performance as a language model. We choose QQP, QNLI, SST-2, and MRPC tasks for our evaluation because they cover all these three areas of GLUE benchmark. QQP is a sentence pair classification task that aims to indicate whether a pair of questions on Quora website is a duplicate or not. QNLI is a sentence pair classification task for predicting whether a question-answer pair is an entailment or not. SST-2 is a single sentence classification task with two annotations, which aims to predict the sentiment of movie reviews. MRPC is a sentence pair classification task, which aims to predict the semantic equivalence between each pair of sentences. We report scores on the development sets of these tasks by fine-tuning on each task.

**Competitors.** We compare the performance of our proposed SᴇɴꜱɪMɪx to the following competitors.

- **BERT pruning.** This method compresses BERT by applying a pruning technique. The method prunes small weights in the embedding and encoder layers of BERT.

- **DistilBERT.** This method uses half of the layers of BERT-base model and applies knowledge distillation during pre-training. DistilBERT consists of an embedding layer and six Transformer encoder layers with the same dimensions as BERT-base.

- **ALBERT.** This method applies parameter sharing and embedding factorization to compress BERT. Specifically, the model uses cross-layer parameter sharing on the Transformer encoder layers and applies matrix factorization on the original embedding layer of BERT.

- **Q8BERT.** This method applies 8-bit quantization to BERT. It quantizes both input and weights to INT8 numerical format and multiplies them using simulated INT8 GEMM.

**Model architecture.** The architecture of SᴇɴꜱɪMɪx is the same as that of BERT except for the type and the number of encoder layers. SᴇɴꜱɪMɪx consists of two types of encoders. One is 8-bit encoder, which applies 8-bit index quantization to all the parameters in the encoder layer

except bias and LayerNorm layers. The other is mixed precision (MP) encoder, which applies 8-bit index quantization to Self-Attention layer and applies 1-bit value quantization to FFN. In our experiments, we choose SENSIMIX (3+6) and SENSIMIX (3+3) as our favorites considering the model size, accuracy, and inference time altogether. SENSIMIX (3+6) represents SENSIMIX with nine layers, three of which are MP encoders and the other six are 8-bit encoders. SENSIMIX (3+3) has six layers, half of which are 8-bit encoders and the other half are MP encoders. The first model has outstanding accuracy with a relatively small model size compared to the competitors. The second model has a better compression rate and inference speed, with slightly decreased but still comparable accuracy.

**Model training.** We train 6 epochs for all tasks (QQP, QNLI, SST-2, and MRPC), and set the initial learning rate to 3e-5, maximum sequence length to 128, and training batch size to 16.

**Model inference.** We set the maximum sequence length to 128 and batch size to 128. We use a single NVIDIA RTX 2080Ti GPU for all the experiments. We conduct experiments in the GPU environment because it is one of the most commonly used hardware in recent deep learning research. The SENSIMIX model is implemented using PyTorch and XNOR GEMM is implemented using PyTorch CUDA extension. We initialize SENSIMIX by a pre-trained BERT model and conduct experiments on the four GLUE benchmark downstream tasks mentioned above. We run each experiment five times and report the average and standard deviation.

## Overall Performance (Q1)

We summarize the accuracy, model size, and inference speed of SENSIMIX along with the competitors. Table 2 and Fig 6 show the overall performance of our proposed SENSIMIX compared to the competitors. We analyze the experimental results from two perspectives: (1) accuracy vs. model size and (2) inference speed.

**Accuracy vs. model size.** We examine the relations of the accuracy and the model size of SENSIMIX compared to the competitors. Fig 6 shows the results for the four GLUE tasks, where x-axis and y-axis denote the model size and the accuracy (f1 score for MRPC), respectively. The percentage (%) of model size is calculated with respect to BERT-base, which means BERT-base has 100% model size. SENSIMIX achieves the best compression rate while maintaining comparable accuracy. Specifically, BERT-base shows the best average accuracy which is 90.9, but it has the largest model size. By contrast, Our SENSIMIX (3+6) achieves a very similar accuracy which is 90.4 but up to 6 times smaller model size compared to BERT-base. The pruning method shows a decent accuracy for the percentage of model size in 20% and 30% while it suffers from a significant accuracy drop when the percentage of model size becomes 10%. However, our SENSIMIX (3+3) shows a much better accuracy compared to the pruning method with the similar model size. DistilBERT gives a good accuracy on the tasks, but it still has a large model size which is 61.2% of BERT-base. ALBERT-large also shows a good accuracy with a small model size, but it has a significant disadvantage that the inference time is much longer than the competitors, as shown in Table 2.

Compared to Q8BERT, SENSIMIX gains benefits in both accuracy and model size. Q8BERT shows a decent trade-off in accuracy and model size, which are 25% of the original model size and 90.2 of average accuracy. However, our SENSIMIX (3+6) reduces the model size to 7.7% of the original model, while achieving an even higher accuracy which is 90.4. Besides, our SENSIMIX (3+3) has half of the model size of Q8BERT with only 0.6 average accuracy degradation. Overall, SENSIMIX gives the best trade-off in terms of accuracy and model size.

**Inference speed.** We assess the inference speed of SENSIMIX and its competitors in Table 3. We set BERT-base as our baseline model. We do not add the BERT pruning method

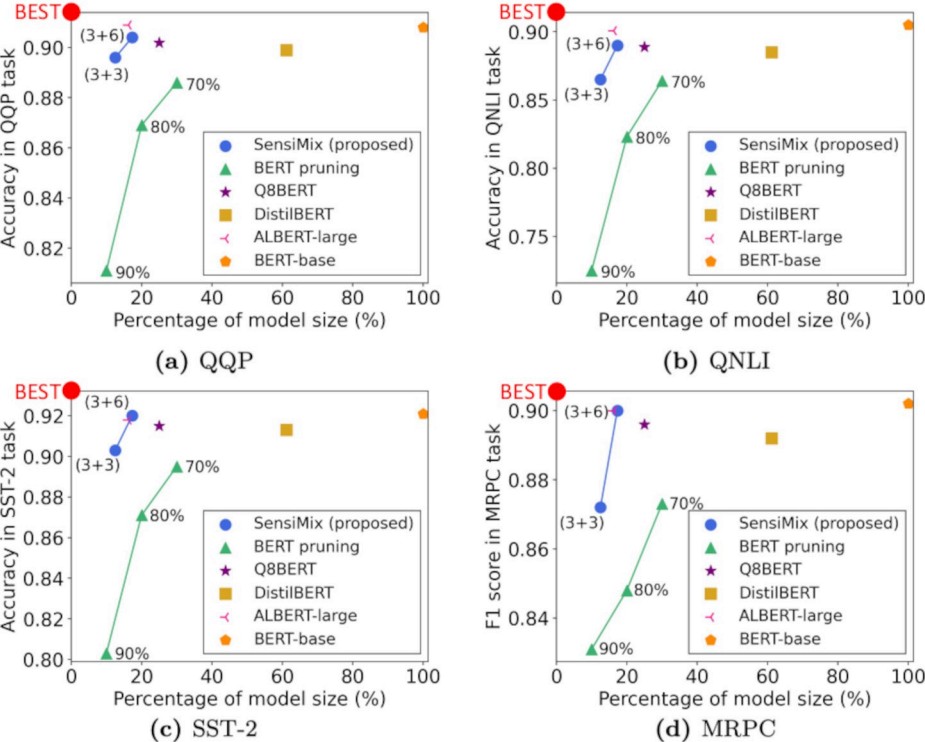

**Fig 6. Accuracy vs. model size for QQP, QNLI, SST-2, and MRPC tasks.** SENSIMIX shows the best trade-off between accuracy and model size. The two points of SENSIMIX represent SENSIMIX (3+3) and SENSIMIX (3+6). The three points of BERT pruning represent the pruning with ratios of 90%, 80%, and 70%.

in the comparison because the pruning method requires special hardware to accelerate the inference speed. We also do not consider Q8BERT as our competitor in inference speed because Q8BERT only simulates the process of INT8 inference by using fake quantization. According to [15], Q8BERT can be accelerated in a special hardware, but SENSIMIX is designed to make inference in the NVIDIA GPU environment.

In Table 3, note that BERT-base takes a long inference time which is 480 seconds, and SENSI-MIX shows the best inference speed among the competitors. ALBERT-large shows about 3× longer inference time than that of BERT-base, since it still consists of 24 parameter-shared encoder layers. DistilBERT achieves a noticeable improvement in inference time which is 248 seconds in average because it has only half of the encoder layers compared to BERT-base. SEN-SIMIX (3+3) takes only 95 seconds to make inference on all the tasks' training sets on average, which is about 5× faster than BERT-base, with only a marginal drop in accuracy.

### Effectiveness of 1-bit training methods (Q2)

We describe through experiments the effectiveness of our proposed three 1-bit quantization-aware training methods, ABWR, PT, and ILF. We choose SENSIMIX (3+3) as our representative model and evaluate the effectiveness of each method by comparing SENSIMIX (3+3) with and without applying each method. Experimental results are summarized in Table 4.

**Effectiveness of Absolute Binary Weight Regularization (ABWR).** Introducing ABWR in the loss function increases the accuracy by 0.4%, 0.7%, 1.5%, and 1.8% in QQP, QNLI, SST-2, and MRPC tasks, respectively, compared to without introducing it, showing an average of

**Table 4. Effectiveness of the three proposed 1-bit quantization-aware training methods.** ABWR, PT, and ILF improve the performance (average score in the GLUE tasks) of SᴇɴꜱɪMɪx by 1.1%, 1.4%, and 1.4%, respectively.

| Methods | Avg. | QQP (Acc) | QNLI (Acc) | SST-2 (Acc) | MRPC (F1) |
|---|---|---|---|---|---|
| **SensiMix** | **88.4** | **89.6±0.2** | **86.5±0.2** | **90.3±0.1** | **87.2±0.2** |
| SᴇɴꜱɪMɪx without ABWR | 87.3 | 89.2±0.2 | 85.8±0.1 | 88.8±0.3 | 85.4±0.4 |
| SᴇɴꜱɪMɪx without PT | 87.0 | 89.1±0.4 | 85.1±0.4 | 88.7±0.3 | 85.2±0.4 |
| SᴇɴꜱɪMɪx without ILF | 87.0 | 89.0±0.3 | 85.2±0.1 | 88.8±0.2 | 85.1±0.5 |

1.1% improvement in accuracy. These consistent results show that ABWR is capable of minimizing the accuracy drop when 1-bit quantization is applied to the model.

**Effectiveness of Prioritized Training (PT).** Applying PT in the training phase improves the accuracy of SᴇɴꜱɪMɪx in QQP, QNLI, SST-2, and MRPC tasks by 0.5%, 1.4%, 1.6%, and 2.0%, respectively. PT improves the accuracy of the model by an average of 1.4% in the four GLUE tasks, which proves the effectiveness of PT.

**Effectiveness of Inverse Layer-wise Fine-tuning (ILF).** ILF enhances the accuracy of SᴇɴꜱɪMɪx in QQP, QNLI, SST-2, and MRPC tasks by 0.6%, 1.3%, 1.5%, and 2.1%, respectively. Overall, ILF increases the accuracy of the model in all the tasks by an average of 1.4%, which validates the effectiveness of our method ILF.

## Sensitivity analysis (Q3)

We mentioned that different parts of BERT have different degree of sensitivity to quantization. We experimentally examine the sensitivity of different parts of the model.

**Sensitivity of Self-Attention layer and FFN.** We compare the sensitivity of Self-Attention layer and FFN in an encoder. For the purpose, we compare two models. The first model, which is SᴇɴꜱɪMɪx, applies 8-bit index quantization to the Self-Attention layer and 1-bit quantization to FFN. The second model applies 1-bit quantization to the Self-Attention layer and 8-bit index quantization to FFN, which is opposite to the original SᴇɴꜱɪMɪx. SᴇɴꜱɪMɪx (3+3) is chosen as the representative model for the study. Table 5 shows that SᴇɴꜱɪMɪx with 1-bit Self-Attention layers shows lower accuracy by an average of 3% compared to SᴇɴꜱɪMɪx with 1-bit FFN. We conclude that the Self-Attention layer is more sensitive to 1-bit quantization than FFN in an encoder.

**Sensitivity of different encoder layers.** We compare the sensitivity of different encoder layers in BERT. For the purpose, we compare the following three models, SᴇɴꜱɪMɪx, SᴇɴꜱɪMɪx-L, and SᴇɴꜱɪMɪx-E. Assuming we use (3+3) model with three 8-bit encoders and three MP encoders, SᴇɴꜱɪMɪx applies MP encoders to the upper three layers (close to the output layer), SᴇɴꜱɪMɪx-L applies MP encoders to the lower three layers, and SᴇɴꜱɪMɪx-E applies MP encoders to the even-numbered layers 2, 4, and 6. Table 6 shows that the original SᴇɴꜱɪMɪx (3+3), which deploys MP encoders to the upper three layers, achieves the best accuracy. This demonstrates

**Table 5. Comparison of the sensitivity of Self-Attention layer and FFN in BERT.** The result indicates that Self-Attention (SA) layer is more sensitive than FFN.

| Methods | Avg. | QQP (Acc) | QNLI (Acc) | SST-2 (Acc) | MRPC (F1) |
|---|---|---|---|---|---|
| **SensiMix 1-bit FFN** | **88.4** | **89.6±0.2** | **86.5±0.2** | **90.3±0.1** | **87.2±0.2** |
| SᴇɴꜱɪMɪx 1-bit SA layer | 85.4 | 86.5±0.2 | 84.5±0.3 | 87.5±0.2 | 83.0±0.1 |

**Table 6. Comparison of the sensitivity of different encoder layers.** SᴇɴsɪMɪx applies MP encoders to the upper three layers, SᴇɴsɪMɪx-L applies MP encoders to the lower three layers, and SᴇɴsɪMɪx-E applies MP encoders to the even-numbered layers 2, 4, and 6. SᴇɴsɪMɪx shows the best accuracy.

| Methods | Avg. | QQP (Acc) | QNLI (Acc) | SST-2 (Acc) | MRPC (F1) |
|---------|------|-----------|------------|-------------|-----------|
| **SensiMix** | **88.4** | **89.6±0.2** | **86.5±0.2** | **90.3±0.1** | **87.2±0.2** |
| SᴇɴsɪMɪx-L | 85.1 | 86.7±0.2 | 84.2±0.4 | 87.0±0.2 | 82.3±0.3 |
| SᴇɴsɪMɪx-E | 85.4 | 87.5±0.2 | 84.2±0.1 | 87.3±0.1 | 82.5±0.3 |

that the encoder layers near the output layer are relatively less sensitive to quantization compared to the other layers.

## Conclusion

We propose SᴇɴsɪMɪx, a novel sensitivity-aware mixed precision quantization method for BERT compression. SᴇɴsɪMɪx quantizes the sensitive and insensitive parts of the model to 8-bit and 1-bit, respectively, to maximize the compression rate and minimize the accuracy drop. For the 8-bit quantization, we exploit 8-bit index quantization that uses 8-bit indices to efficiently represent all weights of each layer of the model, which reduces the model size and keeps the similar accuracy of BERT. For the 1-bit quantization, we propose three novel 1-bit training methods, Absolute Binary Weight Regularization (ABWR), Prioritized Training (PT), and Inverse Layer-wise Fine-tuning (ILF) to minimize the accuracy drop. For fast inference, we apply FP16 general matrix multiplication (GEMM) and XNOR-Count GEMM to the 8-bit and 1-bit parts of the model, respectively. Experiments show that SᴇɴsɪMɪx provides the best compression rate and the fastest inference speed while maintaining the similar accuracy of BERT in four GLUE benchmark tasks. Future works include deploying SᴇɴsɪMɪx on various small devices, comparing to other BERT quantization methods, designing advanced BERT quantization methods, and exploring how to effectively combine genetic algorithms with SᴇɴsɪMɪx that gives better compression, retains more accuracy, and achieves faster inference speed.

## Author Contributions

**Conceptualization:** Tairen Piao, U. Kang.

**Data curation:** Tairen Piao, Ikhyun Cho.

**Formal analysis:** Tairen Piao, Ikhyun Cho.

**Investigation:** Tairen Piao, Ikhyun Cho.

**Methodology:** Tairen Piao.

**Supervision:** U. Kang.

**Validation:** Ikhyun Cho.

**Writing – original draft:** Tairen Piao, U. Kang.

**Writing – review & editing:** Ikhyun Cho.

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
