## [Decision Letter · Decision Letter 0]

5 Oct 2021

PONE-D-21-27639SensiMix: Sensitivity-Aware 8-bit Index & 1-bit Value Mixed Precision Quantization for BERT CompressionPLOS ONE

Dear Dr. Kang,

Thank you for submitting your manuscript to PLOS ONE. After careful consideration, we feel that it has merit but does not fully meet PLOS ONE’s publication criteria as it currently stands. Therefore, we invite you to submit a revised version of the manuscript that addresses the points raised during the review process.

In overall, this is a rich and interesting article. It is well written and described. The authors should further improve it by following all the minor comments provided by the three reviewers to meet PLOS ONE publication criteria.

Please take carefully into account the comments of all the referees for improving the manuscript to meet the required standards by PLOS ONE before resubmitting it to the journal.

We look forward to receiving your revised manuscript.

Kind regards,

Sergio Consoli

Academic Editor

PLOS ONE

Journal Requirements:

 [This work was supported by Institute of Information & Communications Technology Planning & Evaluation (IITP) grant funded by the Korea government (MSIT) (No.2020-0-00894, Flexible and Efficient Model Compression Method for Various Applications and Environments, and No.2017-0-01772, Development of QA systems for Video Story Un derstanding to pass the Video Turing Test). The Institute of Engineering Research and ICT at Seoul National University provided research facilities for this work.]

3. We note you have included a table to which you do not refer in the text of your manuscript. Please ensure that you refer to Table 2 in your text; if accepted, production will need this reference to link the reader to the Table.

4. Please include a copy of Table ?? which you refer to in your text on pages 13 and 14.

Reviewers' comments:

Reviewer's Responses to Questions

**Comments to the Author**

1. Is the manuscript technically sound, and do the data support the conclusions?

Reviewer #1: Yes

Reviewer #2: Yes

Reviewer #3: Yes

2. Has the statistical analysis been performed appropriately and rigorously? 

Reviewer #1: Yes

Reviewer #2: Yes

Reviewer #3: Yes

3. Have the authors made all data underlying the findings in their manuscript fully available?

Reviewer #1: Yes

Reviewer #2: Yes

Reviewer #3: No

4. Is the manuscript presented in an intelligible fashion and written in standard English?

Reviewer #1: Yes

Reviewer #2: Yes

Reviewer #3: No

5. Review Comments to the Author

Reviewer #1: The research presented here is very interesting, and I'm excited to see such nice size and speed gains through smart use of quantization. I feel that the manuscript is mostly clear and concise, however there area few areas I would like to see clarified or elaborated on.

I'm confused by the terms "8-bit index" and "1-bit value". If I am understanding correctly, both 8-bit and 1-bit are quantization strategies on the weights matrices. I'm not sure what the index in "8-bit index" refers to, or why it's different from the value in "1-bit value", other than in bit-width. If the difference between the two strategies is only the amount of quantization (and which layers you apply them to), I would suggest you either name them similarly to reflect this (8-bit value and 1-bit value quantization), or just refer to them as 8-bit quantization and 1-bit quantization.

In table 1, you state that matrix Wldq is an "8-bit index-de-quantized weight matrix of the layer l". However, my understanding is that your de-quantization process takes a matrix of INT8s and (with the min and max) regenerates an approximation of the original matrix of 32FPs. Shouldn't this matrix contain 32-bit values? If not, why not?

This also leads to confusion in 8-bit de-quantization, where the de-quantized values are weight e.g. -1.28, 0.005, 1.00, and 1.27, but Wldq is said to be an 8-bit matrix. How can you store a value of -1.28 inside a INT8?

Line 312 then describes it as an 8-bit clip function, but if the input to alg. 4 was indeed an 8-bit signed integer, the clip function should have no effect. The only way I can make sense of this is if x is actually a FP32, but that doesn't agree with what you've previously written about Wldq.

Line 268: "8-bit index quantization consists of two steps: quantization and de-quantization."

This sentence is very confusing. I understand that both quantization and de-quantization are elaborated on later, but I find it confusing to have step 1 of quantization be its self, and step 2 be undoing its self. I think what you're getting at is something like,

"In order to utilize 8-bit quantization, SensiMix is able to quantize 32FP values to 8-bit int values, and de-quantize the values back to 32-bit.", but elaboration would make this clearer.

I have some issues with the differentiability of your clipping functions. In the 8-bit clip, if the input are FP32 values, values outside of the range (-128,127) will be clipped to -128 and 127. However, the clip function is undifferentiable at this values. Why is this not a problem? You mention in the 1-bit clip section that most values are near zero, and are unlikely to be near the clipping limits, but I don't have a similar kind of intuition about the 8-bit clipping. I would appreciate if you would elaborate about why both clip functions being undifferentiable at their min and max is not a problem (or if it is, explain).

Figure 3. I'm really not sure what this figure is trying to convey. Shouldn't the weights after ABWR be clustered at -1 and 1? To my understanding of ABWR, the left and right subfigures seem swapped; the values before ABWR should be clustered around 0, and afterwards they should be clustered around -1 and 1. Could you explain why Fig 3 shows what it does?

392: Figure 4 seems to show a layer FP32 FFN being converted into a 1-bit FFN after 1 epoch, but that doesn't really agree with your description of adding MP layers to the bottom (I assume the bottom of the existing MP encoder layers, but you don't specify this). Which is correct for ILF?

small details

Line 43: you might want to describe 1-bit quantization here, or cite a description. I assume it quantizes values to -1 or 1, but a reader could easily assume it quantizes it to 0 or 1.

"For the 1-bit quantization, we apply standard 1-bit quantization." This sentence adds nothing, either explain what 1-bit quantization is, or remove this sentence.

Line 146: suggest changing "low-bit numbers" to something like "reduced precision numbers", or "smaller bit-width integers", since low-bit is ambiguous, and to my ears "low-bit" refers to the position of a bit, not the width of an integer.

452: a table link is broken.

520: broken table link.

Reviewer #2: The authors present a method for adaptive weight quantization for the BERT model. The results support the novelty of their method and the paper is well-written and easy to follow. Here are a few comments to help the authors revise the manuscript:

1) Outside the BERT domain there seems to be a body of literature on Mixed Precision Quantization based on the sensitivity of the outputs. e.g. these came up with a simple search: https://ieeexplore.ieee.org/document/9207413, https://arxiv.org/abs/2103.10051

2) There also seems to be more literature on quantization of BERT models e.g. https://arxiv.org/abs/2101.01321, https://arxiv.org/abs/2101.05938, https://web.stanford.edu/class/archive/cs/cs224n/cs224n.1194/reports/custom/15742249.pdf, https://arxiv.org/pdf/2010.07109.pdf which seems to be ignored by the authors.

Could the authors elaborate more on this to stress their novelty?

3) Line 452 and 520, 538 reference to Table is missing.

4) In the questions outlined in Experiments, the section numbering is missing.

Reviewer #3: (1) The term BERT should be expounded first before first time usage.

(2) The introduction was well prepared (well done!)

(3) The section on related works should be expanded.

(4) The section " Related Work " should be " Related Works ".

(5) The proposed method SENSIMIX should be expanded to include algorithmic description in form of flowhart or pseudocode. This is necessary to see the actual operation of the SENSIMIX

(6) The authors did nice work with textual description, but the work should be expanded to include diagramatic illustrations

(7) Is there anyway that Genetic Algorithm (GA) can be used for model reduction in this work ?

(8) Both 8-bit and MP encoders should be described with a flowchart or pseudocode.

(9) How is FB32 matrix in line 276 represented ? (In terms of matrix)

(10) Why the use of the method in Equation 6? Is it better than ADAM method, or SGM ?

(11) In overall, this is a very rich article and well written and described. The authors should include more diagrams and include another section on the description of the computational platform (software) used.

6. PLOS authors have the option to publish the peer review history of their article (what does this mean?). If published, this will include your full peer review and any attached files.

Reviewer #1: No

Reviewer #2: No

Reviewer #3: **Yes: **Oluleye Hezekiah Babatunde

---

## [Author Response · Author response to Decision Letter 0]

15 Nov 2021

1. Response to Reviewer 1 

We would like to thank Reviewer 1 for the insightful and detailed feedback. Comments and how we address them in our paper are summarized below.

[Comment 1] 

• I’m confused by the terms ”8-bit index” and ”1-bit value”. If I am understanding correctly, both 8-bit and 1-bit are quantization strategies on the weights matrices. I’m not sure what the index in ”8-bit index” refers to, or why it’s different from the value in ”1-bit value”, other than in bit-width. If the difference between the two strategies is only the amount of quantization (and which layers you apply them to), I would suggest you either name them similarly to reflect this (8-bit value and 1-bit value quantization), or just refer to them as 8-bit quantization and 1-bit quantization.

[Response to comment 1] 

Q. About the difference between the index quantization and the value quantization.

R. The main difference between the 8-bit index quantization and the 1-bit value quantization depends on whether there is a de-quantization process, in addition to the bit sizes. The 1-bit value quantization does not have a de-quantization process, and all weights are quantized to ±1 and propagated to the next layer directly. However, the 8-bit index quantization has a de-quantization process where FP32 values are first quantized to INT8 indices (we call them indices because they are not the final values that are propagated to the next layer) and then de-quantized back to FP32 values before propagated to the next layer. In the revised paper, we added a precise description of the difference between the value and the index quantization to lines 343-345.

[Comment 2] 

• In table 1, you state that matrix Wldq is an ”8-bit index-de-quantized weight matrix of the layer l”. However, my understanding is that your de-quantization process takes a matrix of INT8s and (with the min and max) regenerates an approximation of the original matrix of 32FPs. Shouldn’t this matrix contain 32-bit values? If not, why not? This also leads to confusion in 8-bit de-quantization, where the dequantized values are weight e.g. -1.28, 0.005, 1.00, and 1.27, but Wldq is said to be an 8-bit matrix. How can you store a value of -1.28 inside a INT8? Line 312 then describes it as an 8-bit clip function, but if the input to alg. 4 was indeed an 8-bit signed integer, the clip function should have no effect. The only way I can make sense of this is if x is actually a FP32, but that doesn’t agree with what you’ve previously written about Wldq.

[Response to comment 2] 

Q. About the definition of the matrix Wldq 

R. It is right that Wldq is the de-quantized FP32 weight matrix which contains FP32 values, and we stated this matrix as ”8-bit index-de-quantized weight matrix of the layer l”, which might bring misunderstandings to readers. In the revised paper, we revised the definitions of Wlq and Wldq in Table 1 to make the statement more clear.

[Comment 3] 

• Line 268: ”8-bit index quantization consists of two steps: quantization and de-quantization.” This sentence is very confusing. I understand that both quantization and de-quantization are elaborated on later, but I find it confusing to have step 1 of quantization be its self, and step 2 be undoing its self. I think what you’re getting at is something like, ”In order to utilize 8-bit quantization, SensiMix is able to quantize 32FP values to 8-bit int values, and de-quantize the values back to 32-bit.”, but elaboration would make this clearer. 

[Response to comment 3] 

Q. About the elaboration of the process of the 8-bit index quantization.

R. It is right that the quantization step quantizes FP32 values to INT8 values (indices), and the de-quantization step de-quantizes INT8 values back to FP32 values. In the revised paper, we added a precise elaboration about this part to lines 279-281. 

[Comment 4] 

• I have some issues with the differentiability of your clipping functions. In the 8-bit clip, if the input are FP32 values, values outside of the range (-128,127) will be clipped to -128 and 127. However, the clip function is undifferentiable at this values. Why is this not a problem? You mention in the 1-bit clip section that most values are near zero, and are unlikely to be near the clipping limits, but I don’t have a similar kind of intuition about the 8-bit clipping. I would appreciate if you would elaborate about why both clip functions being undifferentiable at their min and max is not a problem (or if it is, explain).

[Response to comment 4]

Q1. About the intuition of the 8-bit clip function .

R1. We use the 8-bit clip function to replace the 8-bit round function in the training process to solve the zero gradient problem of the 8-bit round function. The derivative of the 8-bit round function is equal to zero in almost all ranges which disables the training of the model using gradient descent-based training methods, while replacing it by the 8-bit clip function enables the model training.

Q2. About the problem that values outside of the range (-128,127).

 R2. In the 8-bit index quantization, FP32 weights in a weight matrix are divided into 256 ranges which are temporarily stored in INT8 numbers (indices), and then we use these INT8 numbers, the minimum weight, and the maximum weight to approximate the original FP32 weight matrix (de-quantization step). Note that in Equation 1, all original FP32 weight values are mapped into range [-128, 127], which means all FP32 weights are updated. In the revised paper, we added statements about this in lines 335-337 to make it more clear.

[Comment 5] 

• Figure 3. I’m really not sure what this figure is trying to convey. Shouldn’t the weights after ABWR be clustered at -1 and 1? To my understanding of ABWR, the left and right subfigures seem swapped; the values before ABWR should be clustered around 0, and afterwards they should be clustered around -1 and 1. Could you explain why Fig 3 shows what it does? 

[Response to comment 5] 

R. The order of the figures in Figure 3 was wrong, we corrected the order of the figures in the revised paper. 

[Comment 6] 

• 392: Figure 4 seems to show a layer FP32 FFN being converted into a 1-bit FFN after 1 epoch, but that doesn’t really agree with your description of adding MP layers to the bottom (I assume the bottom of the existing MP encoder layers, but you don’t specify this). Which is correct for ILF? 

[Response to comment 6] 

R. It is right that the ”bottom” represents the bottom of the existing MP encoder layers. To make the statement more clear, we revised the lines 407, 408, 410, and 411 which give a more precise description of ILF in the revised paper.

[Comment 7] 

• Line 43: you might want to describe 1-bit quantization here, or cite a description. I assume it quantizes values to -1 or 1, but a reader could easily assume it quantizes it to 0 or 1. ”For the 1-bit quantization, we apply standard 1-bit quantization.” This sentence adds nothing, either explain what 1-bit quantization is, or remove this sentence. 

[Response to comment 7] 

R. We removed the sentence in the revised paper. 

[Comment 8] 

• Line146: suggest changing ”low-bit numbers”to something like”reduced precision numbers”, or ”smaller bit-width integers”, since low-bit is ambiguous, and to my ears ”low-bit” refers to the position of a bit, not the width of an integer. 

[Response to comment 8] 

R. We changed the words ”low-bit numbers” to ”smaller bit-width integers” in the revised paper.

[Comment 9] 

• 452: a table link is broken. 520: broken table link.

[Response to comment 9] 

R. We fixed the broken reference links in the revised paper.

2. Response to Reviewer 2 

We would like to thank Reviewer 2 for the high-quality review and constructive comments. Comments and how we address them in our paper are summarized below.

 [Comment 1] 

• 1) Outside the BERT domain there seems to be a body of literature on Mixed Precision Quantization based on the sensitivity of the outputs. e.g. these came up with a simple search: [5, 4] 2) There also seems to be more literature on quantization of BER T models e.g. [3, 2, 1, 6]. Could the authors elaborate more on this to stress their novelty?

[Response to comment 1] 

Q1. About the contribution and novelty of our paper.

 R1. Our main claim is to compress the BERT model to a small one while maintaining its accuracy, and we have the following contributions: 

–  We discover the sensitivity of different modules of BERT. We demonstrate that the encoders close to the input layer are more sensitive than those near to the output layer in BERT, and Self-Attention layer is more sensitive than feed-forward network (FFN) in an encoder. 

–  We effectively compress the model considering the sensitivity of different modules. We compress the sensitive parts of BERT using 8-bit index quantization and insensitive parts using 1-bit value quantization, which maximizes the compression rate while minimizing the accuracy degradation caused by quantizing sensitive modules to low bits. 

–  We propose three 1-bit quantization-aware training methods: Absolute Binary Weight Regularization (ABWR), Prioritized Training (PT), and Inverse Layer-wise Fine-tuning (ILF). They overcome the challenges of training 1-bit quantization method, improving the accuracy of the model. 

–  We introduce a fast inference strategy of SENSIMIX on an NVIDIA GPU. We apply FP16 general matrix multiplication (GEMM) and XNOR-Count GEMM to replace the original GEMM for the 8-bit and the 1-bit quantization parts of the model, respectively, to achieve fast inference speed.

Q2. About comparisons with other mixed precision quantization methods. 

R2. We compared our method to Q8BERT which is an effective quantization method for BERT. Other methods are excluded for the following reasons. First, there are difficulties when comparing our method to quantization methods designed on special hardware because the performance changes when the hardware environment changes. Second, methods that are designed for other models (MLP, CNN, and RNN) are not directly applicable to BERT, such as [5] and [4]. However, we agree that comparing to other mixed precision BERT quantization methods is one of the next steps. We added the comment in line 635 in the revised paper.

[Comment 2] 

• Line 452 and 520, 538 reference to Table is missing. 

[Response comment 2] 

R. We fixed the broken reference links in the revised paper.

[Comment 3] 

• In the questions outlined in Experiments, the section numbering is missing.

[Response to comment 3] 

R. In the revised paper, we removed the section numbers in the experimental questions and added the corresponding question numbers to the title of each section.

3. Response to Reviewer 3 

We would like to thank Reviewer 3 for the insightful and detailed comments. Comments and our responses are summarized below. 

[Comment 1] 

• The term BERT should be expounded first before first time usage. 

[Response to comment 1] 

R. In the revised paper, we added an introduction of BERT in the first subsection of Related Works (lines 84-92) to expound the term BERT.

[Comment 2] 

• The introduction was well prepared (well done!) 

[Response to comment 2] 

R. Thank you for your kind comment. 

[Comment 3] 

• The section on related works should be expanded. 

[Response to comment 3] 

R. We added an introduction of BERT to the first subsection of Related Works (lines 84-92). This subsection expands the related works and provides a clear elaboration of BERT to readers.

[Comment 4] 

• The section ”Related Work” should be ”Related Works”. 

[Response to comment 4] 

R. We revised the term ”Related Work” to ”Related Works” in the revised paper.

[Comment 5] 

• The proposed method SENSIMIX should be expanded to include algorithmic description in form of flowchart or pseudocode. This is necessary to see the actual operation of the SENSIMIX 

[Response to comment 5] 

R. In the revised paper, we added a pseudocode algorithm of SENSIMIX (Algorithm 1) in Proposed Method.

[Comment 6] 

• The authors did nice work with textual description, but the work should be expanded to include diagramatic illustrations 

[Response to comment 6] 

R. In the revised paper, we added a pseudocode algorithm of SENSIMIX (Algorithm 1) and a figure (Fig 3) on the process of the 8-bit index quantization in Proposed Method to expand diagramatic illustrations of our method.

[Comment 7] 

• Is there anyway that Genetic Algorithm (GA) can be used for model reduction in this work ? 

[Response to comment 7] 

R. We have not combined Genetic Algorithms with our method. However, we added it to our future work to lines 636-637 in the revised paper.

[Comment 8] 

• Both 8-bit and MP encoders should be described with a flowchart or pseudocode.

[Response to comment 8] 

R. In the revised paper, we added a pseudocode algorithm of SENSIMIX (Algorithm 1) to Proposed Method which includes the algorithms of 8-bit and MP encoders. Furthermore, we added a figure (Fig 3) on the process of the 8-bit index quantization to Proposed Method to give a diagramatic explanation of the 8-bit encoder.

[Comment 9] 

• How is FB32 matrix in line 276 represented ? (In terms of matrix) 

[Response to comment 9] 

R. The FP32 matrix in your comment is the full-precision weight matrix of each layer in the model (corresponding to Wlfp in Table 1). In the revised paper, we gave a more clear elaboration to lines 289-290.

[Comment 10] 

• Why the use of the method in Equation 6? Is it better than ADAM method, or SGM? 

[Response to comment 10] 

R. Equation6 is a basic training rule to solve the zero gradient problem of the binary function; other gradient descent-based update rules (e.g., Adam) can be used as well. To make the statement more clear, we added an explanation of Equation 6 to lines 367-369 in the revised paper.

[Comment 11] 

• In overall, this is a very rich article and well written and described. The authors should include more diagrams and include another section on the description of the computational platform (software) used. 

[Response to comment 11] 

R. Thank you for your valuable comments. In the revised paper, we added a pseudocode algorithm of SENSIMIX (Algorithm 1) and an illustration figure of the process of the 8-bit index quantization (Fig 3) in Proposed Method, and we added a description of our software environment to lines 530-531.

References 

[1]  Fan, C.: Quantized transformer

[2]  Jin, J., Liang, C., Wu, T., Zou, L., Gan, Z.: KDLSQ-BERT: A quantized bert combining knowledge distillation with learned step size quantization. CoRR abs/2101.05938 (2021). URL https://arxiv.org/abs/2101.05938  

[3]  Kim, S., Gholami, A., Yao, Z., Mahoney, M.W., Keutzer, K.: I-bert: Integer-only bert quantization (2021)  

[4]  Lee, D., Cho, M., Lee, S., Song, J., Choi, C.: Data-free mixed-precision quantization using novel sensitivity  metric. CoRR abs/2103.10051 (2021). URL https://arxiv.org/abs/2103.10051  

[5]  Yuan, Y., Chen, C., Hu, X., Peng, S.: Evoq: Mixed precision quantization of dnns via sensitivity guided evolutionary search. In: 2020 International Joint Conference on Neural Networks (IJCNN), pp. 1–8 (2020). DOI 10.1109/IJCNN48605.2020.9207413 

[6]  Zhao, Z., Liu, Y., Chen, L., Liu, Q., Ma, R., Yu, K.: An investigation on different underlying quantization schemes for pre-trained language models (2020)

---

## [Decision Letter · Decision Letter 1]

3 Jan 2022

PONE-D-21-27639R1SensiMix: Sensitivity-Aware 8-bit Index & 1-bit Value Mixed Precision Quantization for BERT CompressionPLOS ONE

Dear Dr. Kang,

Thank you for submitting your manuscript to PLOS ONE. After careful consideration, we feel that it has merit but does not fully meet PLOS ONE’s publication criteria as it currently stands. Therefore, we invite you to submit a revised version of the manuscript that addresses the points raised during the review process.

The paper has improved evidently and the contents are worth of interest for the community. There are however still points to be addressed before the manuscript can reach an acceptable standard level for being published. 

In particular make sure to address the comment by R2 who raised a concern about the background section and experimental evaluation relative to other methods in the literature, making thus in discussion the novelty of the proposed method.

If an additional experimental evaluation is not possible, it should be at least reported a full literature review in the background section related to quantization methods for neural nets. Such a literature review should explain the main differences with each method, explaining in case why it was not reported in the comparison by the authors. It should be stressed then also why the proposed SensiMix method is novel relative to these methods.

Please take carefully into account the comments of all the referees for improving the manuscript to meet PLOS ONE standards before resubmitting it to the journal.

We look forward to receiving your revised manuscript.

Kind regards,

Sergio Consoli

Academic Editor

PLOS ONE

Journal Requirements:

Reviewers' comments:

Reviewer's Responses to Questions

**Comments to the Author**

1. If the authors have adequately addressed your comments raised in a previous round of review and you feel that this manuscript is now acceptable for publication, you may indicate that here to bypass the “Comments to the Author” section, enter your conflict of interest statement in the “Confidential to Editor” section, and submit your "Accept" recommendation.

Reviewer #2: (No Response)

Reviewer #4: All comments have been addressed

2. Is the manuscript technically sound, and do the data support the conclusions?

Reviewer #2: Partly

Reviewer #4: Yes

3. Has the statistical analysis been performed appropriately and rigorously? 

Reviewer #2: Yes

Reviewer #4: Yes

4. Have the authors made all data underlying the findings in their manuscript fully available?

Reviewer #2: Yes

Reviewer #4: Yes

5. Is the manuscript presented in an intelligible fashion and written in standard English?

Reviewer #2: Yes

Reviewer #4: Yes

6. Review Comments to the Author

Reviewer #2: Unfortunately the authors have not addressed my concerns with regards to comparison with other quantization methods, making it very difficult to judge the actual novelty of the paper. If an experimental evaluation is not possible, I think there should be at least a full literature review in the background section in quantization methods for neural nets. Such a literature review should explain the differences with each one, why it wasn't used for comparison, and why it SensiMix is novel.

Reviewer #4: The authors propose an acceleration of the BERT family of models using quantization techniques. Although the contribution is not difficult to pursue, the results are promising in practice. Therefore, I believe that the paper has merit to be accepted in a multidisciplinary journal such as Plos One.

I suggest introducing and discussing recent references on how to increase the computational efficiency of BERT from different perspectives.

Some examples are

BERT Pre-training Acceleration Algorithm Based on MASK Mechanism

Hardware Acceleration of Fully Quantized BERT for Efficient Natural Language Processing

Plug-Tagger: A Pluggable Sequence Labeling Framework Using Language Models

A Comprehensive Survey on Training Acceleration for Large Machine Learning Models in IoTs

7. PLOS authors have the option to publish the peer review history of their article (what does this mean?). If published, this will include your full peer review and any attached files.

Reviewer #2: No

Reviewer #4: No

---

## [Author Response · Author response to Decision Letter 1]

31 Jan 2022

We tried our best to reflect all the reviewers' comments in our revised manuscript. Thank you for the meaningful comments.

---

## [Decision Letter · Decision Letter 2]

7 Mar 2022

SensiMix: Sensitivity-Aware 8-bit Index & 1-bit Value Mixed Precision Quantization for BERT Compression

PONE-D-21-27639R2

Dear Dr. Kang,

We’re pleased to inform you that your manuscript has been judged scientifically suitable for publication and will be formally accepted for publication once it meets all outstanding technical requirements.

Kind regards,

Sergio Consoli

Academic Editor

PLOS ONE

Additional Editor Comments (optional):

Reviewers' comments:

Reviewer's Responses to Questions

**Comments to the Author**

1. If the authors have adequately addressed your comments raised in a previous round of review and you feel that this manuscript is now acceptable for publication, you may indicate that here to bypass the “Comments to the Author” section, enter your conflict of interest statement in the “Confidential to Editor” section, and submit your "Accept" recommendation.

Reviewer #2: All comments have been addressed

Reviewer #4: All comments have been addressed

2. Is the manuscript technically sound, and do the data support the conclusions?

Reviewer #2: Partly

Reviewer #4: Yes

3. Has the statistical analysis been performed appropriately and rigorously? 

Reviewer #2: Yes

Reviewer #4: Yes

4. Have the authors made all data underlying the findings in their manuscript fully available?

Reviewer #2: Yes

Reviewer #4: Yes

5. Is the manuscript presented in an intelligible fashion and written in standard English?

Reviewer #2: Yes

Reviewer #4: Yes

6. Review Comments to the Author

Reviewer #2: The authors have added a detailed background section referring to similar works and addressing my earlier concerns.

Reviewer #4: The authors added my suggestions and improve the manuscript in this review round. Congratulations on your work!

7. PLOS authors have the option to publish the peer review history of their article (what does this mean?). If published, this will include your full peer review and any attached files.

Reviewer #2: No

Reviewer #4: No

---

## [Editor Report · Acceptance letter]

8 Apr 2022

PONE-D-21-27639R2 

SensiMix: Sensitivity-Aware 8-bit Index & 1-bit Value Mixed Precision Quantization for BERT Compression 

Dear Dr. Kang:

I'm pleased to inform you that your manuscript has been deemed suitable for publication in PLOS ONE. Congratulations! Your manuscript is now with our production department. 

Kind regards, 

on behalf of

Dr. Sergio Consoli 

Academic Editor

PLOS ONE